# Experimental Investigation on the Discharge of Pollutants from Tunnel Fires

**Lihua Zhai [1,2], Zhongxing Nong [1], Guanhong He [1], Baochao Xie [3], Zhisheng Xu [3] and Jiaming Zhao [3,\*]** 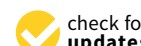

1   Guangzhou Metro Design and Research Institute Co., Ltd., Guangzhou 510000, China; zhailihua@dtsjy.com (L.Z.); nongxingzhong@dtsjy.com (Z.N.); heguanhong@dtsjy.com (G.H.)
2   Key Laboratory of Road and Traffic Engineering of Ministry of Education, Tongji University, Shanghai 201804, China
3   Institute of Disaster Prevention Science and Safety Technology, Central South University, Changsha 410075, China; xiebaochao@csu.edu.cn (B.X.); xuzhisheng@csu.edu.cn (Z.X.)
\*   Correspondence: zhaojiaming93@csu.edu.cn; Tel.: +86-731-8265-6625

**Abstract:** Many pollutants are generated during tunnel fires, such as smoke and toxic gases. How to control the smoke generated by tunnel fires was focused on in this paper. A series of experiments were carried out in a 1:10 model tunnel with dimensions of 6.0 m × 1.0 m × 0.7 m. The purpose was to investigate the smoke layer thickness and the heat exhaust coefficient of the tunnel mechanical smoke exhaust mode under longitudinal wind. Ethanol was employed as fuel, and the heat release rates were set to be 10.6 kW, 18.6 kW, and 31.9 kW. The exhaust velocity was 0.32–3.16 m/s, and the longitudinal velocity was 0–0.47 m/s. The temperature profile in the tunnel was measured, and the buoyant flow stratification regime was visualized by a laser sheet. The results showed that the longitudinal ventilation leads to a secondary stratification of the smoke flow. In the ceiling extract tunnel under longitudinal ventilation, considering the research results of the smoke layer height and the heat exhaust coefficient, a better scheme for fire-producing pollutants was that an exhaust velocity of 1.26–2.21 m/s (corresponding to the actual velocity of 4.0–7.0 m/s) should be used. The longitudinal velocity should be 0.16–0.32 m/s (corresponding to the actual velocity of 0.5–1.0 m/s).

**Keywords:** tunnel fire; pollutant control; smoke layer height; heat exhaust coefficient

## 1. Introduction

The terrain environment in different regions was usually distinct. The complex terrain environment leads to inconvenient traffic conditions in the region. People use tunnels to alleviate this problem [1]. As of April 12, 2019, there were 17,738 highway tunnels in China with a length of 17.2361 million meters, an increase of 1509 places and 1.951 million meters, including 1058 long tunnels with a length of 0.47066 million meters, an increase of 156 places and 693,400 m, and 4315 long tunnels with a total length of 0.74218 million meters, an increase of 474 places and 822,500 m [2].

The tunnel is a long, narrow, semi-enclosed space. When a fire breaks out in a tunnel, the consequences would be even more serious if it is not controlled properly. According to incomplete statistics, more than 16 major tunnel fire accidents occurred in China between 2000 and 2016 [3]. In these fire cases, the smoke was the biggest threat to personnel safety and structural facilities. In order to control the smoke, predecessors invented a variety of smoke exhaust methods, such as a full transversal smoke extraction system, ceiling smoke extraction, longitudinal ventilation, vertical shaft smoke exhaust, combined smoke exhaust, and so on.

In order to investigate the longitudinal ventilation mode, Memorial tunnel studied this mode early, the results showed that the control effect of smoke in the tunnel was not good because there was almost

no longitudinal wind in the tunnel. This phenomenon was the balance between air supplement and smoke exhaust in the full transverse smoke exhaust system. It was illustrated that longitudinal wind played a key role in smoke control [4]. Ventilation alms arrangement was studied in the model-scale tunnel of "air + helium" and it was found that the installation of the fan at the top was better than the side suspension to exhaust smoke [5]. Then, the distribution of temperature field and the length of smoke recirculation in the tunnel under normal and blocked scenarios were studied extensively [6–8].

In order to investigate the ceiling smoke extraction, the Zhejiang Provincial Transportation Planning and Design Research Institute of China and Central South University jointly carried out scientific research to jointly tackle the key technologies of central smoke extraction, structure fire resistance, and the concentrated smoke exhaust schemes in independent smoke exhaust pipes of highway tunnels [9]. They used a combined mode of tunnel operation shaft with jet fan, and the concept of central smoke extraction with independent smoke exhaust pipes at the top in tunnel fires [10]. Based on the practical projects such as QianJiang tunnel and Hong Kong–Zhuhai–Macao tunnel, the temperature distribution, smoke emissions, and other technical parameters of different smoke emission models were obtained through numerical simulation [11,12]. The influence of induced wind speed on the control effect of the central smoke extraction tunnel was investigated through numerical simulation [13,14]. In recent years, the extent of backflow length downstream of the dampers, the "confinement velocity" (suppressing the back-layering flow downwind of the vents), and the stability of the smoke stratification was investigated in the transverse ventilation system [15–17].

In order to investigate the vertical shaft smoke exhaust, the influence of vertical wind, vertical shaft layout, strong environmental wind, and vertical shaft structure on the air entraining-suction model was investigated through scale-model test, numerical simulation, and theoretical analysis [18–20]. Given the possibility of smoke emission in the vertical shaft tunnel, the prediction model of smoke back-layering length was established [21]. Then, the relationship between the horizontal driving force and longitudinal inertial force during the smoke movement was investigated; the critical criterion *Ri* which was used to determine whether the smoke outlet was plug holing was proposed [22]. The distribution of smoke in railway tunnels under natural ventilation mode was investigated in a 1:15 model-scale tunnel, and the corresponding empirical formula was derived. The formula could be used to predict ceiling temperature distribution and smoke exhaust [23]. The tunnel fire smoke exhaust system under the coupling of longitudinal ventilation and shaft mechanical smoke exhaust was investigated through a scaled (1:5) model; this combined method was beneficial to smoke exhaust and evacuation [24].

In addition, the influence of induced ventilation on the smoke exhaust effect under the mode of centralized exhaust was studied extensively [25–27]. The variation of system heat efficiency under different semi-transverse smoke exhaust modes, different opening modes of smoke exhaust vents, and different ventilation forms was studied [28,29]. Some scholars also investigated the fire spread of materials [30].

This paper mainly investigated the combined model of the longitudinal ventilation + ceiling smoke extraction, the diagram shown in Figure 1. Based on the analysis of the buoyancy plume structure, the smoke layer height, and the exhaust efficiency, the characteristics and ventilation scheme of smoke exhaust were determined.

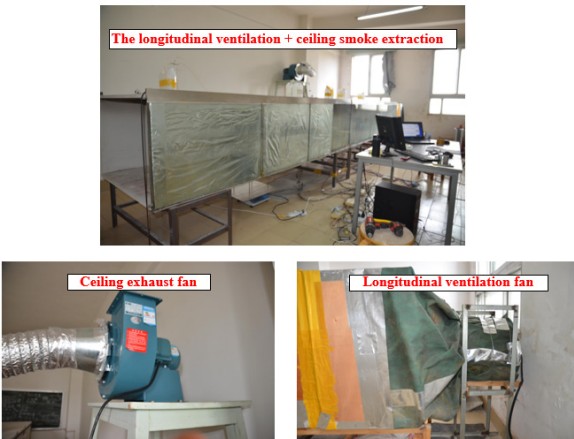

**Figure 1.** Schematic diagram of the longitudinal ventilation + ceiling smoke exhaust mode.

## 2. Experimental Setup

### 2.1. Experimental Rigs

Froude scaling was a common scaling method in tunnel smoke experiments [31]. According to the Froude modeling, the heat release rate (HRR), test time, flow rates, the energy content, and mass are scaled. A reduced-scale tunnel (1:10 scale) was used in this paper, as shown in Figure 2, which length, width, and height were 6.0 m, 1.0 m, and 0.7 m, respectively. This model tunnel was introduced in our previous research [32]. It was constructed by using a fireproof steel frame structure, fireproof slab, and fireproof glass. The ambient temperature during the experiment was 13 °C to 20 °C.

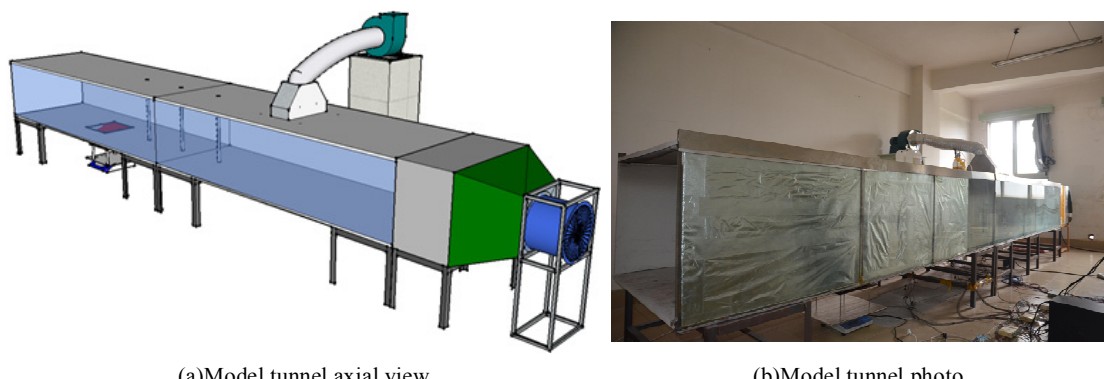

(a)Model tunnel axial view                    (b)Model tunnel photo

**Figure 2.** Experimental setup. ((**a**) Model tunnel axial view; (**b**) Model tunnel photo)

The experimental tunnel was equipped with a mechanical exhaust system, and the exhaust vent was located on the tunnel ceiling. The exhaust vent was rectangular, the size was 0.4 m × 0.125 m (3 m away from the fire source). The ventilation system was arranged at one end of the tunnel, including the axial fan and the steady flow section. The exhaust velocity in the test model was 0–3.16 m/s, which corresponds to the exhaust velocity of 0–10 m/s in the actual tunnel, and the longitudinal wind speed in the test model was 0–0.47 m/s, which was equivalent to the longitudinal wind speed of 0–1.5 m/s in the actual tunnel.

### 2.2. Measurement

The data of the fuel mass change, smoke temperature, and smoke flow velocity were determined in the experiment. As shown in Figure 3, 39 K-type thermocouples were used to collect the temperature along the longitudinal central section of the tunnel and the vertical temperature in the one-dimensional horizontal spread stage.

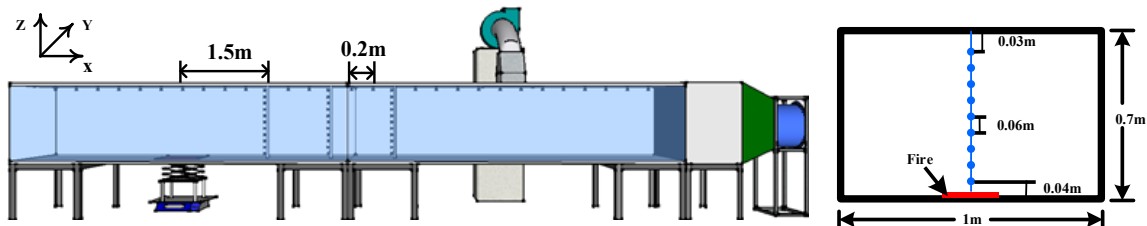

**Figure 3.** Schematic diagrams for the measurement positions.

Generally, the maximum uncertainty error of the thermocouple was lower than 6% [33]. In this work, the uncertainty of the measured temperature was determined to be approximately 6.5% through comparison of two repeated experiments running under the same conditions. The distance between the thermocouple and the top of the tunnel was 0.03 m, and the distance between the thermocouples on the thermocouple tree was 0.06 m.

The mass data were measured by an electronic balance placed at the bottom of the tunnel. A camera was placed on the side of the observation window to capture the buoyancy flow stratification. And three velocity measuring points and three temperature measuring points were arranged in the exhaust vent. An armored K-type thermocouple with a diameter of 0.1 mm was adopted to be the thermocouple, the diameter of the K-type thermocouple was very small, and the response time was about 0.01 s, which satisfied the experimental requirements in this study. The range of wind speed acquisition sensor was 0–25 Pa.

*2.3. Experimental Procedure*

The longitudinal wind was generated by a fan set downstream of the mechanical exhaust vent. The mechanical exhaust was produced by a centrifugal fan that was connected to the top of the tunnel model. They could be controlled by adjusting the frequency converter. In this paper, the actual longitudinal velocity of tunnel engineering was 0 m/s, 0.5 m/s, 1.0 m/s and 1.5 m/s, and the mechanical smoke exhaust velocity was 1 m/s, 2 m/s, 4 m/s, 7 m/s and 10 m/s. According to the Froude scaling, the longitudinal ventilation velocity of the test model (1:10) was 0 m/s, 0.16 m/s and 0.32 m/s, and the mechanical smoke exhaust velocity of the test model (1:10) was 0.32 m/s, 1.26 m/s, 2.16 m/s and 3.16 m/s. The longitudinal ventilation velocity and the mechanical exhaust smoke velocity were the velocity without fire source. In this paper, the parameters of the scale model were utilized.

The size of the oil pan was $15 \times 15$ cm, $20 \times 20$ cm, $25 \times 25$ cm, the fuel was anhydrous ethanol. The average mass loss rate in the stable section of the oil pan of different sizes was 0.0004 kg/s, 0.0007 kg/s, and 0.0012 kg/s. The heat release rate was calculated by the following formula:

$$\dot{Q} = \eta \times m_f \times \Delta H \tag{1}$$

where $\eta$ is the combustion efficiency, $m_f$ is the mass loss, and $\Delta H$ is the combustion heat of absolute anhydrous ethanol and is 26.78 kJ/g, and the combustion efficiency of absolute ethanol is 0.994, The selection of parameters has been mentioned in previous studies [32]. The heat release rates of the fire source were calculated as 10.6 kW, 18.6 kW, and 31.9 kW.

Under the conditions of different smoke exhaust velocity and different longitudinal velocity, 60 groups of test conditions were carried out. Table 1 was an experimental test condition.

**Table 1.** Experimental test condition.

| Test Series | Heat Release Rate (kW) | Exhaust Velocity (m/s) | Longitudinal Velocity (m/s) |
|---|---|---|---|
| V1–V20 | 10.6 | 0.32, 0.63, 1.26, 2.21, 3.16 | 0, 0.16, 0.32, 0.47 |
| V21–V40 | 18.6 | 0.32, 0.63, 1.26, 2.21, 3.16 | 0, 0.16, 0.32, 0.47 |
| V41–V60 | 31.9 | 0.32, 0.63, 1.26, 2.21, 3.16 | 0, 0.16, 0.32, 0.47 |

## 3. Results and Discussion

*Buoyant Flow Stratification Regimes Visualizations*

Analyzing the buoyant flow structure was beneficial to investigate the influence of airflow on smoke. Taking the heat release rate of 31.9 kW, the exhaust velocity of 0.32 m/s, 1.26 m/s, 3.16 m/s, the longitudinal velocity of 0 m/s, 0.16 m/s as examples, the buoyant flow structure was visualized by a laser sheet with the cold smoke test. The upper green was hot smoke layer and the lower was fresh air.

As shown in Figure 4, as the exhaust velocity increases, the depression was occurred under the exhaust vent. The phenomenon was mentioned in previous studies on plug-holing [22,34]. This phenomenon became more pronounced as the exhaust wind speed increases.

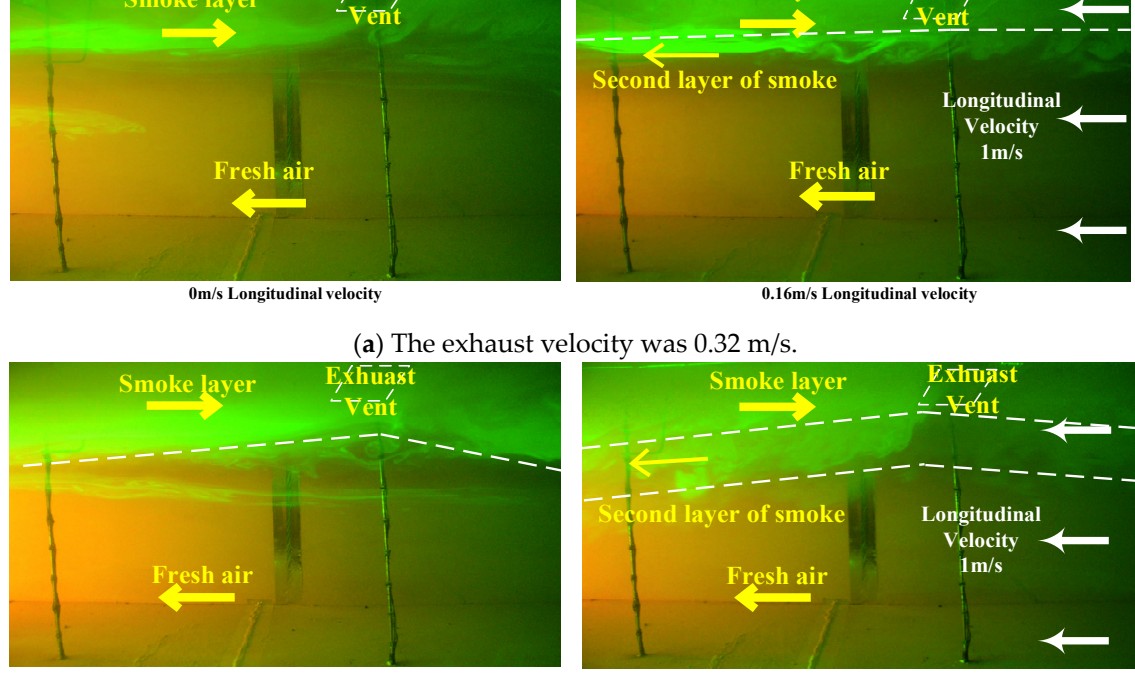

(**a**) The exhaust velocity was 0.32 m/s.

(**b**) The exhaust velocity was 0.63 m/s.

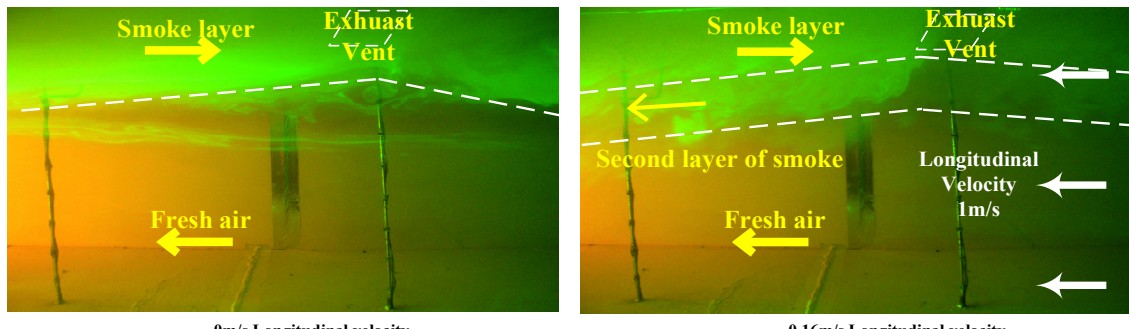

(**c**) The exhaust velocity was 3.16 m/s.

**Figure 4.** 31.9 kW configurations of buoyant flow stratification. ((**a**) The exhaust velocity was 0.32 m/s; (**b**) The exhaust velocity was 0.63 m/s; (**c**) The exhaust velocity was 3.16 m/s)

In this experiment, it was also observed that the smoke movement was more pronounced in the downstream of the vent than the upstream. Because of the coupling effect of the longitudinal velocity and the smoke exhaust velocity, the smoke entrainment downstream of the exhaust hole increases.

Noteworthy, under the action of longitudinal wind, the back-layering of smoke formed a new stratification under the effect of thermal buoyancy. In addition, the air entrapment phenomenon existed on the interface of the two layers, which reduced the depression area below the smoke vent to a certain extent.

As shown in Figure 5, according to the phenomenon observed in the experiment, the smoke movement process was summarized into four sections: (a) the stable section, in which the mechanical smoke exhaust and longitudinal velocity were small; (b) the secondary stratification section, in which the mechanical smoke exhaust velocity was small and the longitudinal velocity was large; (c) the concave section, the mechanical smoke exhaust velocity was relatively large and the longitudinal velocity was relatively low; (d) the complex moving section, both the mechanical smoke exhaust velocity and the longitudinal velocity were relatively high.

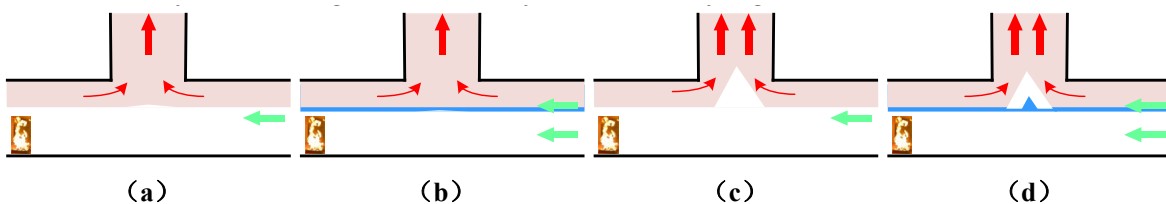

(a)           (b)           (c)           (d)

**Figure 5.** Smoke movement at different stages. ((**a**) the stable section; (**b**) the secondary stratification section; (**c**) the concave section; (**d**) the complex moving section)

## 4. Smoke Layer Height Changes

The finite-current motion in stably stratified fluids and turbulent shear flow in stratified fluids was mentioned [35], with amplitude motion defined as *l*:

$$\frac{\partial^2 l}{\partial t^2} = \frac{g \partial \rho}{\partial z} l \tag{2}$$

The concept of buoyancy frequency for thermal stratification, and the buoyancy frequency refers to the rate of change of buoyancy in the vertical direction proposed [36], the formula is:

$$N_l = (-g \frac{\partial \rho_s}{\partial z} / \rho_a)^{\frac{1}{2}} \tag{3}$$

Usually, the fire plume could be always considered as the ideal gas ($\rho_s T_s = \rho_a T_a$) [37], then the Equation (3) becomes:

$$N_l = (-g T_a \frac{\partial (1/T(z))}{\partial z})^{\frac{1}{2}} \tag{4}$$

$N_L$ is the value of the buoyancy frequency in the vertical direction, $T_a$ is the ambient air temperature, $g$ is the gravitational acceleration, $T(z)$ is the vertical temperature distribution function, $\rho_a$ is the air density, and $\rho_s$ is the smoke density.

An example is shown in Figure 6.

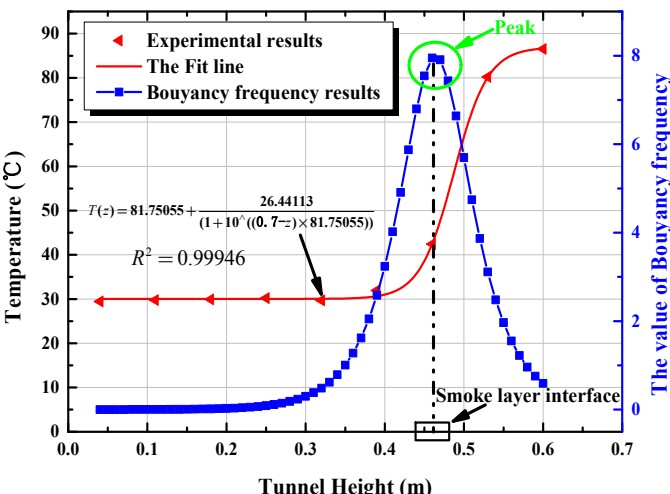

**Figure 6.** The curve of temperature upstream the exhaust vent and $N_L$ values under $HRR$ = 10.6 kW, $V_e$ = 0.32 m/s, $V_L$ = 0.16 m/s.

The changes in the smoke layer height upstream of the smoke vent under different fire scenes are shown in Figure 7.

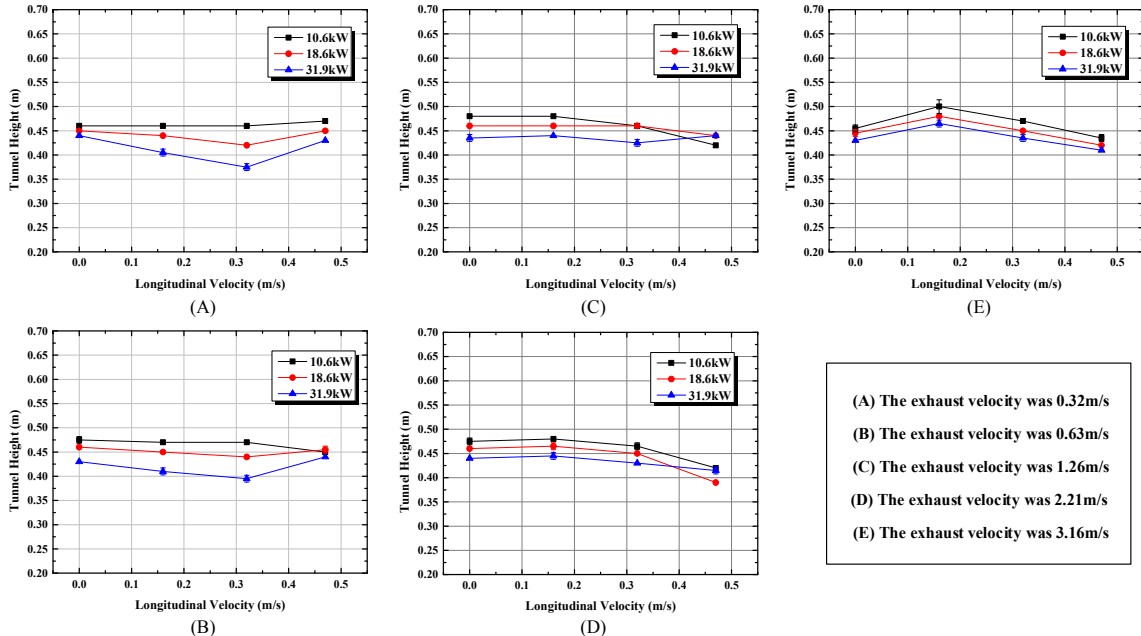

**Figure 7.** Smoke layer height change. ((**A**) The exhaust velocity was 0.32 m/s; (**B**) The exhaust velocity was 0.63 m/s; (**C**) The exhaust velocity was 1.26 m/s; (**D**) The exhaust velocity was 2.21 m/s; (**E**) The exhaust velocity was 3.16 m/s)

As the longitudinal velocity increases, for Figure 7A, when the smoke exhaust velocity was 0.32 m/s, the smoke layer height of the three heat release rates first decreases and then increases slightly; as for Figure 7B, when the smoke exhaust velocity was 0.63 m/s and the heat release rate was 10.6 kW, the smoke layer height was basically unchanged. When the heat release rate was 18.6 kW and 31.9 kW, the smoke layer height was slightly decreased.

For Figure 7C, when the smoke exhaust velocity was 1.26 m/s and the heat release rate was 10.6 kW and 18.6 kW, the smoke layer height decreases slightly, the heat release rate was 31.9 kW, the smoke layer height was basically unchanged; for Figure 7D, when the smoke exhaust velocity was 2.21 m/s, the variation rule of smoke layer thickness was the same as when the smoke exhaust velocity

was 1.26 m/s; as for Figure 7E, the smoke layer thickness of the three heat release rates had the same variation rule, first increasing and then decreasing. When the longitudinal velocity was 0.16 m/s, the smoke layer thickness was the thinnest.

In conclusion, under different conditions of smoke exhaust velocity, the smoke layer thickness in the upstream of the smoke exhaust vent did not change significantly, and it remained within the range of 0.45 m ± 0.5 m, mainly due to the influence of heat release rate. Under the same conditions of smoke exhaust velocity, different longitudinal velocity influences the smoke layer thickness upstream of the smoke exhaust vent.

*Heat Exhaust Coefficient Change*

Yi Liang studied the heat exhaust coefficient of transversal smoke extraction systems in a tunnel under fire [29]. Tanaka studied the efficiency of exhausting heat in the smoke of the longitudinal smoke temperature distribution during a fire in a road tunnel with vertical shafts [38,39]. According to the thermodynamics theorem and the theoretical research foundation of predecessors, the heat carried by the smoke discharged from the smoke vent could be expressed by the enthalpy value flowing through the section of the smoke vent, the formula is:

$$\dot{Q}_E = c_p \rho_s v_s A \Delta T_s \tag{5}$$

The heat exhaust coefficient refers to the ratio of the total heat carried by the smoke from the smoke vent to the fire source, the formula is:

$$\delta = \frac{\dot{Q}_E}{\dot{Q}} \tag{6}$$

The conditions with different heat exhaust coefficients were shown in Figure 8.

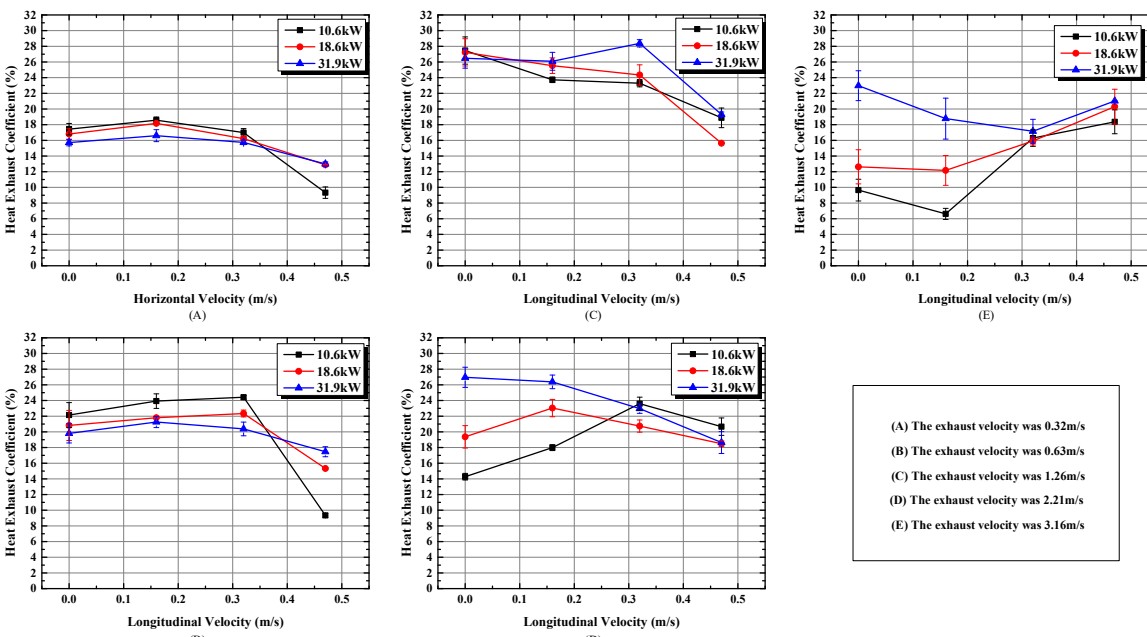

**Figure 8.** Heat exhaust coefficient. ((**A**) The exhaust velocity was 0.32 m/s; (**B**) The exhaust velocity was 0.63 m/s; (**C**) The exhaust velocity was 1.26 m/s; (**D**) The exhaust velocity was 2.21 m/s; (**E**) The exhaust velocity was 3.16 m/s)

As for Figure 8A, when the smoke exhaust velocity was 0.32 m/s, the changing trend of the three heat release rates was the same, and the heat exhaust coefficient increased slightly and then decreased. When the heat release rate was 10.6 kW, the longitudinal wind had the most obvious influence on the heat exhaust coefficient, and the declining trend of the heat exhaust coefficient was greater than the other two heat release rates. As for Figure 8B, when the exhaust velocity was 0.63 m/s, the changing trend of the heat exhaust coefficient was roughly the same as that in Figure 8A, and the decreasing trend of the heat release rate was more significant when it was 10.6 kW. As for Figure 8C, when the exhaust velocity was 1.26 m/s when the longitudinal velocity increased from 0 m/s to 0.32 m/s, there was no obvious change in the heat exhaust coefficient. When the longitudinal velocity increased to 0.47 m/s, the heat exhaust coefficient was shown as a downward trend. As for Figure 8D, when the exhaust velocity was 3.16 m/s, the heat release rate was 31.9 kW. In the working condition, with the increase of longitudinal velocity, the heat exhaust coefficient was slowly decreased, and the declining trend became obvious. Under the conditions of 10.6 kW and 18.6 kW, with the increase of longitudinal velocity, the heat exhaust coefficient first increased and then decreased. As for Figure 8E, under the three conditions of the heat release rate, the changing trend of heat exhaust coefficient was the same. With the increase of longitudinal velocity, it was first decreased and then increased.

According to Figure 8, the coupling effect of mechanical exhaust velocity and longitudinal velocity on the smoke exhaust system has a certain impact. When the longitudinal velocity was 0 m/s, the mechanical exhaust velocity was 0.32 m/s, 0.63 m/s, and 1.26 m/s, and the difference in heat exhaust coefficient was not obvious. When the mechanical exhaust velocity was 2.21 m/s and 3.16 m/s, the heat exhaust coefficient varied widely. Just like the observed phenomenon, the smoke vent was penetrated, resulting in a decrease in the heat exhaust coefficient of 10.6 kW and 18.6 kW. In the same longitudinal velocity, the increase of the exhaust velocity would lead to the heat exhaust coefficient first being increased and then decreased, because there was a plug-holing at the smoke exhaust vent. In this process, the lower longitudinal velocity (0.16–0.32 m/s) would increase the heat exhaust coefficient to some extent, while the higher smoke exhaust velocity (0.32–0.46 m/s) would decrease the heat exhaust coefficient to some degree.

## 5. Conclusions

In this paper, the discharge of pollutants from tunnel fires was investigated by the scale-model. Major findings include the following:

(1) Secondary stratification of smoke caused by longitudinal wind was observed in experimental phenomena.

(2) At the interface of the two smoke layers, there were both vortex and air entrainment, and the thickness of the overall smoke layer became larger. At the same time, the increase of the exhaust velocity was observed, and the area of the recessed area below the exhaust vent was enlarged, and the longitudinal wind reduced the area of the recessed area at the exhaust vent to a certain extent.

(3) In the ceiling extract tunnel under longitudinal ventilation, considering the research results of the smoke layer height and the heat exhaust coefficient, a better scheme for fire-producing pollutants was that the exhaust velocity should be 1.26–2.21 m/s (corresponding to the actual velocity of 4.0–7.0 m/s). The longitudinal velocity should be 0.16–0.32 m/s (corresponding to the actual velocity of 0.5–1.0 m/s).

This investigation lacks factors such as the distance from the fire source in the exhaust vent, the area of the exhaust vent, and the size of the tunnel section in the smoke extraction scheme. In order to perfect the ideas of this paper, the author and other related researchers would then investigate the factors which are not involved in the team's project—new 1:20 tunnel (20 m, 50 m) and the complex tunnel model.

**Author Contributions:** Conceptualization, L.Z., J.Z. and Z.X.; methodology, J.Z., Z.N., G.H. and B.X.; modeling, Z.X., J.Z., and B.X.; validation, J.Z., Z.N. and G.H.; formal analysis, L.Z. and J.Z.; investigation, L.Z., J.Z., J.Z. and G.H.; writing—original draft preparation, L.Z. and J.Z.; writing—review and editing, L.Z. and J.Z.; visualization, Z.N. and G.H.; project administration, L.Z., Z.N., G.H. and Z.X. All authors have read and agreed to the published version of the manuscript.

**Funding:** This research was funded by [the National Key R&D Program of China] grant number [No. 2017YFB1201204]; and the project of Guangzhou Metro Rail Transit Line 18 and 22 Special Disaster Prevention and Rescue.

**Conflicts of Interest:** The authors declare no conflict of interest.

## Nomenclature

| | | | |
|---|---|---|---|
| $\dot{Q}$ | heat release rate (kW) | | **Greek symbols** |
| $T_a$ | ambient air temperature (°C) | $l$ | oscillating distance |
| $T_i$ | smoke layer interface temperature (°C) | $\Delta$ | deviation property |
| $T_{max}$ | maximum temperature of vertical distribution (°C) | $P$ | density (kg/m$^3$) |
| $N_L$ | smoke layer height (m) | $\delta$ | partial derivative |
| $H$ | tunnel height (m) | | **Subscripts** |
| $T(Z)$ | vertical temperature distribution function | $S$ | smoke |
| $g$ | gravitational acceleration (m/s$^2$) | $A$ | ambient air |
| $\Delta H$ | combustion heat of absolute anhydrous ethanol (kJ/g) | $I$ | interface |
| | | $P$ | plume |
| | | $F$ | fire |

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
