# Peer review of "Experimental Investigation on the Discharge of Pollutants from Tunnel Fires"

_sustainability, doi:10.3390/su12051817_

Round 1
Reviewer 1 Report
The paper describes the experimetal tests of the discharge of pullutans
from tunnel fires. The tests have been implemented in a very carefull
way and moreover the experimental results have been well highlighted.
Notwithstanding, the manuscript, for this reviewer seems to lack of a paragraph
relating to the discussion of methods of control of these phenomena.
Other issues are the following:
(1) improve the quality of the equations;
(2) the conclusion are very poor. The last sentence is not conclused.
For these reasons, the paper, before of a its whole acceptance,
has to be improved following the issues raised by this reviewer.
Author Response
Response to review comments (sustainability-711280)
We appreciate very much for the positive reviews and helpful suggestions. Below are the responses to each of the raised point and description of the modifications that have been implemented in the revision. On the marked copy of the revision, response is highlighted with red color.
Reviewer #1
The paper describes the experimetal tests of the discharge of pullutans from tunnel fires. The tests have been implemented in a very carefull way and moreover the experimental results have been well highlighted. Notwithstanding, the manuscript, for this reviewer seems to lack of a paragraph relating to the discussion of methods of control of these phenomena.
Response:
Thank you for the comment.
We had changed the language and added concluding paragraphs.
Lines 164-172: As shown in Figure 4, according to the phenomenon observed in the experiment, the smoke movement process was summarized into four sections: (a) stable section, in which the mechanical smoke exhaust and longitudinal velocity were small; (b) secondary stratification section, in which the mechanical smoke exhaust velocity was small and the longitudinal velocity was large; (c) the concave section, the mechanical smoke exhaust velocity was relatively large and the longitudinal velocity was relatively low; (d) the complex moving section, both the mechanical smoke exhaust velocity and the longitudinal velocity were relatively high.
Lines 205-209: In conclusion, under different of conditions smoke exhaust velocity, the smoke layer thickness in the upstream of the smoke exhaust vent did not change significantly. And it remained within the range of 0.45m±0.5m, mainly due to the influence of heat release rate. Under the same conditions of smoke exhaust velocity, different longitudinal velocity influences the smoke layer thickness upstream of the smoke exhaust vent.
Lines 242-251: According to Figure 7, the coupling effect of mechanical exhaust velocity and longitudinal velocity on the smoke exhaust system has a certain impact. When the longitudinal velocity was 0m/s, the mechanical exhaust velocity was 0.32m/s, 0.63m/s, and 1.26m/s, and the difference in heat exhaust coefficient was not obvious. When the mechanical exhaust velocity was 2.21m/s and 3.16m/s, the heat exhaust coefficient varied widely. Just like the observed phenomenon, the smoke vent was penetrated, resulting in a decrease in the heat exhaust coefficient of 10.6kW and 18.6kW. In the same longitudinal velocity, the increase of the exhaust velocity would lead to the heat exhaust coefficient was first increased and then decreased, because there was a plug-holing at the smoke exhaust vent. In this process, the lower longitudinal velocity (0.16-0.32m/s) would increase the heat exhaust coefficient to some extent, while the higher smoke exhaust velocity (0.32-0.46m/s) would decrease the heat exhaust coefficient to some degree.
Other issues are the following:
(1) improve the quality of the equations;
Response:
Thank you for the comment.
We have modified and improved formula quality.
(2) the conclusion are very poor. The last sentence is not conclused.
Response:
Thanks for your good suggestion.
We had perfected the content of the conclusion.
Lines 255-272: In this paper, the discharge of pollutants from tunnel fires were investigated by the scale-model. Major findings include:
(1)Secondary stratification of smoke caused by longitudinal wind was observed in experimental phenomena.
(2)At the interface of the two smoke layers, there were both vortex and air entrainment, and the thickness of the overall smoke layer became larger. At the same time, the increase of the exhaust velocity was observed, and the area of the recessed area below the exhaust vent was enlarged, and the longitudinal wind reduced the area of the recessed area at the exhaust vent to a certain extent.
(3)In the ceiling extract tunnel under longitudinal ventilation, considering the research results of the smoke layer height and the heat exhaust coefficient, a better

Reviewer 2 Report
The authors discuss a study involving pollutant dispersion in a tunnel fire. The manuscript seems hastily-written, has significant English language flaws, and is of overall low quality in terms of presentation. Additionally, the topic is fairly specialized and the authors have not made efforts to clarify specific concepts or terminology for a reader who is not familiar with the subject matter. Sustainability is a broader impact journal and this manuscript is significantly more specialized.
Specific concerns:
In the nomenclature table, the symbols are completely unaligned and look very messy. The Introduction is very poorly-written, both in structure and English language (grammar, organization, etc.). It needs to be rewritten with significant changes. Line 31: What is the "Director's Tunnel" and why is it important? Lines 33-34: This sentence should not stand by itself, it looks very strange as it is now. Lines 36-37: This assertion needs a supporting citation. Lines 38-40: For readers with limited knowledge of tunnel ventilation, it is critical to provide more explanation or citations. Lines 86-92: There are no citations to support the assertions listed. Overall, both in the Introduction and Methods section, the authors must try to explain the concepts to non-specialists in the field. Reverse ventilation, for example, is not described and I had to read on my own to understand what is meant by this. In Figure 3 what do the different colors mean? This is not clear at all. Section 4 should be listed within the Results section. It is strange to see the authors include a period "." between Figure and the figure number (i.e. Figure.5). In Figure 5, the y-axes of all subplots could be standardized to make the comparison easier.Author Response
Response to review comments (sustainability-711280)
We appreciate very much for the positive reviews and helpful suggestions. Below are the responses to each of the raised point and description of the modifications that have been implemented in the revision. On the marked copy of the revision, response is highlighted with red color.
Reviewer #2
The authors discuss a study involving pollutant dispersion in a tunnel fire. The manuscript seems hastily-written, has significant English language flaws, and is of overall low quality in terms of presentation. Additionally, the topic is fairly specialized and the authors have not made efforts to clarify specific concepts or terminology for a reader who is not familiar with the subject matter. Sustainability is a broader impact journal and this manuscript is significantly more specialized.
Specific concerns:
In the nomenclature table, the symbols are completely unaligned and look very messy. The Introduction is very poorly-written, both in structure and English language (grammar, organization, etc.). It needs to be rewritten with significant changes.
Response:
Thank you for the comment.
We had modified the nomenclature table. The manuscript was thoroughly checked and proofread. Some errors and typos were corrected. These modifications have been highlighted with red color in the revised version.
Line 31: What is the "Director's Tunnel" and why is it important?
Response:
Thank you for the comment.
We checked and proofread, and modified our presentation.
Lines 29-32: As of April 12, 2019, there were 17,738 highway tunnels in China with a length of 17.2361 million meters, an increase of 1,509 places and 1.951 million meters, including 1,058 long tunnels with a length of 0.47066 million meters, an increase of 156 places and 693,400 meters, and 4,315 long tunnels with a total length of 0.74218 million meters, an increase of 474 places and 822,500 meters
Lines 33-34: This sentence should not stand by itself, it looks very strange as it is now.
Response:
Thank you for the comment.
We checked and proofread, and modified our presentation.
Lines 34-35: The tunnel is a long, narrow, semi-enclosed space. When a fire breaks out in a tunnel, the consequences would be even more serious if it is not controlled properly.
Lines 36-37: This assertion needs a supporting citation.
Response:
Thank you for the comment.
We had added supporting citations.
Lines 35-36: According to incomplete statistics, more than 16 major tunnel fire accidents occurred in China between 2000 and 2016 [3].
[3] Ren R., Zhou H., Hu Z., He S.Y., Wang X.L. Statistical analysis of fire accidents in Chinese highway tunnels 2000–2016. Tun. Under. Sp. Tech, 2019, 83, 452-440.
Lines 38-40: For readers with limited knowledge of tunnel ventilation, it is critical to provide more explanation or citations.
Response:
Thanks for your good suggestion.
We deleted the expression here, modified the expression and added diagrams at the end of the introduction.
Lines 81-82: This paper mainly investigated the combined model of the longitudinal ventilation + ceiling smoke extraction, the diagram was shown in Figure 1.
Lines 86-92: There are no citations to support the assertions listed. Overall, both in the Introduction and Methods section, the authors must try to explain the concepts to non-specialists in the field. Reverse ventilation, for example, is not described and I had to read on my own to understand what is meant by this.
Response:
Thanks for your good suggestion.
In order to facilitate the reader's understanding, we had modified the expression. We used the noun " the longitudinal ventilation + ceiling smoke exhaust " and added a figure to illustrate it.
Lines 81-82: This paper mainly investigated the combined model of the longitudinal ventilation + ceiling smoke extraction, the diagram was shown in Figure 1.
In Figure 3 what do the different colors mean? This is not clear at all.
Response:
Thanks for your good suggestion.
The markings were shown in Figure 3 and Section 3.1. The upper green was hot smoke layer and the lower was fresh air.
Lines 144-146: the buoyant flow structure was visualized by a laser sheet with the cold smoke test. The upper green was hot smoke layer and the lower was fresh air.
Section 4 should be listed within the Results section.
Response:
Thank you for the comment.
We had changed the language and added concluding paragraphs.
Lines 242-251: According to Figure 7, the coupling effect of mechanical exhaust velocity and longitudinal velocity on the smoke exhaust system has a certain impact. When the longitudinal velocity was 0m/s, the mechanical exhaust velocity was 0.32m/s, 0.63m/s, and 1.26m/s, and the difference in heat exhaust coefficient was not obvious. When the mechanical exhaust velocity was 2.21m/s and 3.16m/s, the heat exhaust coefficient varied widely. Just like the observed phenomenon, the smoke vent was penetrated, resulting in a decrease in the heat exhaust coefficient of 10.6kW and 18.6kW. In the same longitudinal velocity, the increase of the exhaust velocity would lead to the heat exhaust coefficient was first increased and then decreased, because there was a plug-holing at the smoke exhaust vent. In this process, the lower longitudinal velocity (0.16-0.32m/s) would increase the heat exhaust coefficient to some extent, while the higher smoke exhaust velocity (0.32-0.46m/s) would decrease the heat exhaust coefficient to some degree.
It is strange to see the authors include a period "." between Figure and the figure number (i.e. Figure.5).
Response:
Thanks for your good suggestion.
We had checked and modified.
In Figure 5, the y-axes of all subplots could be standardized to make the comparison easier.
Response:
Thank you for the comment.
We had normalized the y-axis in the figure.
Round 2
Reviewer 1 Report
The manuscript has been significantly improved and so it is
advisable for a possible publication in this journal